# Resilience, Occupational Stress, Job Satisfaction, and Intention to Leave the Organization among Nurses and Midwives during the COVID-19 Pandemic

**DOI:** 10.3390/ijerph19116826

**Published:** 2022-06-02

**Authors:** Andrzej Piotrowski, Ewa Sygit-Kowalkowska, Ole Boe, Samir Rawat

**Affiliations:** 1Institute of Psychology, University of Gdańsk, 80-309 Gdansk, Poland; 2Department of Psychology, Kazimierz Wielki University, 85-867 Bydgoszcz, Poland; esygit@ukw.edu.pl; 3Department of Business, Strategy and Political Sciences, University of South-Eastern Norway, 3045 Drammen, Norway; ole.boe@usn.no; 4Institute of Psychology, Oslo New University College, 0456 Oslo, Norway; 5Military MIND Academy, Pune 411060, India; samtanktrooper@yahoo.com

**Keywords:** nurses, midwives, resilience, occupational stress, job satisfaction, intention to leave, the COVID-19 pandemic

## Abstract

The current study on the intention to leave the organization among nurses and midwives aligns with the broader direction of research on the consequences of demanding jobs. This is particularly important in the context of the COVID-19 pandemic, which began in 2020 and is ongoing. The aim of the current study was to identify the levels of intention to leave the organization and job satisfaction in a sample of 390 Polish nurses and midwives. A multiple stepwise linear regression was carried out to establish which variables are predictors of job satisfaction and intention to leave the organization. The following measures were used in the study: Nurses’ Occupational Stressor Scale, The Brief Resilience Coping Scale, The Turnover Intention Scale, The Job Satisfaction Scale, and an occupational questionnaire (number of workplaces, weekly number of evening and night shifts, working at a unit dedicated to treating COVID-19, working as a supervisor/executive). The current study showed that almost 25% of the sample reported high turnover intention, and a similar proportion reported low job satisfaction. Resilience was related to nurses’ job satisfaction. In the predictive models for job satisfaction, the organizational factor of the number of workplaces was significant (positively related), while job experience was a negative predictor of intention to leave. The practical implications of the results and the need to continue research on this topic are also discussed.

## 1. Introduction

In Poland, nurses and midwives both belong to a single group of mid-level medical professionals. They are organized under a shared regulatory college and informational-educational centers, and they share the same code of ethics, together with other legislative regulations that treat these two professions jointly.

Nursing is indicated as a particularly demanding profession, and a career in nursing involves heightened occupational stress. Its intensity is significantly higher in this group compared to the general working population [1]. The stressful situations specific to nursing include the high pace of work, work requiring the physical transferring of patients, long hours, night shifts, working in several institutions, overtime, highly emotional engagement with the patients and their families, overburdening with responsibilities and shiftwork, insufficient pay, and stressful workplace interactions [2,3]. Such situations can negatively impact nurses’ health and job satisfaction, as well as contribute to the intention to leave the organization [4].

Nurses and midwives are sought after on the job market. This trend is only increasing, and systemic issues are failing to address it systematically. The demand for nurses is especially increasing. This is due to the increasing numbers of aging and disabled persons in the population, which was observable already before the pandemic [5]. Analyses carried out in Poland show that the proportion between nurses’ turnover (also due to retirement) and the entrance of new nurses into the profession will be negative in the coming years and that in the years 2008–2020, it has been estimated to number 66 thousand people [6].

Thus far in the course of the SARS-CoV-2 pandemic, healthcare professionals have experienced severe work overload, exposure to infection, lack of appropriate personal protective equipment, moral dilemmas, unpleasant interactions at work, discrimination, despair, and isolation from their families [7]. Healthcare professionals risked their health and lives, resulting in fear of infection and occupational stress [8]. The latter phenomenon stems from a lack of balance between job demands and the workers’ capabilities [9]. Nurses have experienced traumatic stress in the context of their patients’ significantly increased mortality rate. Additionally, undergoing severe infections themselves led to a culmination of traumatic stress [10]. It has been observed that healthcare professionals experienced such stress due to becoming infected with the COVID-19 disease compared to other groups. The International Council of Nurses reports that by October 2020, 1500 nurses have died due to COVID-19 across 44 countries. This occupational group is also indicated as having the highest infection rate at 38.6% (n = 10,706) [11,12]. Several particularly troubling phenomena have intensified during the pandemic and which negatively impact medical professionals’ mental health include stigmatization, occupational burnout, depression, and experiencing physical violence [13].

The need for studying the characteristics of mid-level healthcare professionals’ jobs is evidenced by results showing that overburden with work, stress, and dangerous workplace conditions are the main factors influencing the quality of provided care [14]. Occupational stress negatively impacts job satisfaction and is linked to cognitive dysfunctions, irritability, and intention to leave the organization [15]. The literature also points to occupational stress being a mediator between employee optimism and risk of occupational burnout and between mindfulness and depression or anxiety [16].


**Intention to leave the organization**


The intention to leave the organization is characterized by a declared intent by the employee to leave the place of work. It can involve reduced work engagement. Empirical studies have identified the intention to leave as a predictor of actual resigning [17]. Intention to leave is also related to experiencing occupational burnout and lack of support from one’s supervisors [18]. Simultaneously, the intention to leave is negatively correlated with loyalty to supervisors [19].

Studies carried out by the European project NEXT among nurses in 10 countries showed that Polish nurses displayed the highest levels of loyalty and attachment to their profession. Around 64% of them have indicated never thinking about leaving nursing in the past year. In other countries, the proportion of nurses who have never considered quitting varied from 59% (The Netherlands) to 34% (Great Britain) [20]. Polish nurses aged under 40 and working in intensive medical care declared intention to leave more often, and the most commonly indicated reason was low pay. Over 40% of Polish nurses work in two organizations [21].

The intention to leave the organization in the context of the SARS-CoV-2 pandemic has been studied in many European and Asian countries [22]. These studies show that the pandemic has significantly increased the intention among nurses compared to the pre-pandemic period. Li et al. showed that one in six healthcare professionals reported the intention to leave during the pandemic [23]. Fear of COVID-19 is also positively related to the intention to resign among student nurses [24]. Other studies also show a relationship between anxiety, including fear of infection, and nurses’ intention to leave the organization during the COVID-19 pandemic [25].


**Job satisfaction**


Job satisfaction is described as a positive orientation toward one’s job [26]. Its level is closely related to the employee’s motivation and productivity, and it is indicated as the most significant predictor of occupational stress [27]. The factors constituting job satisfaction include, among others, one’s career and status as an employee, health and well-being at work, personal development, relationships between work and personal life, mental and physical effort, interpersonal relationships at work, or job security, using and recognizing coworkers’ abilities, the nature of one’s job, the range of responsibilities, and the promotion system [28].

Research shows that nurses’ job satisfaction is significantly related to patients’ safety, quality of care, and employee loyalty. Low job satisfaction is related to the overall level of missed nursing care [29]. Simultaneously, low job satisfaction is also related to a lack of communication in the employee team and a lack of supervisor support [30]. Low job satisfaction among nurses can lead to more frequent absences and lower quality of patient care [31]. Among the countries included in the European NEXT project, Polish nurses reported the lowest job satisfaction [20]. Other data indicate that low and very low job satisfaction was indicated by 75% of Polish nurses and, for comparison, only 16% of Norwegian nurses [32].

During the pandemic, a significant part of the surveyed nurses indicated a lack of job satisfaction together with missed care and perceived lack of support [33]. Nurses with high job satisfaction were characterized by higher professional commitment [34]. In turn, nurses’ low job satisfaction during the SARS-CoV-2 pandemic was related to role conflicts, workload, and psychosomatic problems [35].

Thus far, job satisfaction has been identified as a mediator between occupational stress and intention to leave such that the higher the job satisfaction, the lower the occupational stress and intention among long-term care nurses [36].


**Resilience**


Lazarus and Folkman indicated that in order for a situation to cause stress for an individual, it has to be appraised as stressful; that is, the recourse demands must exceed the individual’s subjectively rated capabilities [37]. One of the most important resources (protective factors) in coping with stress is resilience.

Most researchers, including the creators of this term, treat resilience as a relatively stable disposition determining the process of flexible adaptation to the constantly changing demands of life [38]. Research confirms the link between resilience and job-related stress, burnout, and well-being [39].

Among nurses, resilience prevents avoidance-related and depressive behaviors [40]. Results have also confirmed the relationship between resilience and the experience of stress. Nurses characterized by higher resilience experience lower levels of general stress and occupational burnout [41]. Highly resilient nurses more frequently possess such personal resources as optimism or active coping skills, which could be developed through therapeutic interventions and training [42]. Resilience has been identified as a partial mediator between stress and life satisfaction in a sample of medical students [43]. Nurses report a medium level of stress and a high level of resilience, which has a significant and negative correlation with stress, anxiety, and depression. In the relationship between stress and mental health, resilience constitutes a protective factor [44].

The aim of the current study was to identify the levels of intention to leave and job satisfaction in a sample of Polish nurses and midwives during the pandemic. The following variables were analyzed in relation to the development of intention to resign and job satisfaction:–Occupational stress;–Resilience;–Organizational factors: number of workplaces, weekly number of evening and night shifts, working at a unit dedicated to treating COVID-19 or treating patients diagnosed with COVID-19, working as a supervisor/executive. The model of the tested variables is shown in Figure 1.

The following research questions were put forward:What are the levels of intention to leave the organization and job satisfaction in the studied sample?Do resilience, organizational factors, and occupational stress levels differentiate intention to leave and job satisfaction levels?Which of the variables included in the study are significant predictors of intention to leave and job satisfaction in the studied sample?

## 2. Materials and Methods

### 2.1. Study Collects

Data collection took place from 22 November 2020 to 21 March 2021. Participation in the study was voluntary and anonymous. The first step involved establishing cooperation with nursing/midwife organizations. Information about the study was shared, together with an invitation to share it with members of the contacted organizations. The information included the study aims and measures. This information was sent to all Regional Councils of Nurses and Midwives in Poland. The councils shared a Google Forms (Google LLC, Mountain View, CA, USA) link to the study on their websites. This way, a sample of nurses and midwives was collected from all over Poland. The following criteria for inclusion were adopted: informed consent to participate, being a professional nurse or midwife, and being professionally active during the COVID-19 pandemic. It was assumed that the current health status, including a potential COVID-19 diagnosis, would not be used as an inclusion criterion. All participants were informed about the goal of the research.

### 2.2. Measurements

The following measures were used in the study:

Nurses’ Occupational Stressor Scale (NOSS) was developed by Chen et al. [45]. The authors have obtained written permission to produce a Polish translation of this scale. The NOSS is comprised of 21 items forming 9 scales: Work Demands, 3 items (e.g., “I have to bear negative sentiment from patients or their relatives.”), Work-Family Conflict, 3 items (e.g., “The burden of work makes it difficult for me to undertake my personal chores and/or engage in hobbies.”), Insufficient Support from Coworkers or Caregivers (e.g., “I feel stressed because primary caregivers do not execute their tasks appropriately.”), Workplace Violence and Bullying, 1 item (“I feel stressed due to psychological abuse such as threats, discrimination, bullying, and harassment.”), Organizational Issues, 3 items (e.g., “The on-call system affects my life.”), Occupational Hazards, 2 items (e.g., “I feel stressed considering that my patients might be have contagious diseases such as SARS or AIDS”), Difficulty Taking Leave, 2 items (e.g., “I cannot ask for leaves for household emergencies.”), Powerlessness, 2 items (e.g., “It upsets me if patients’ conditions do not improve.”), and Unmet Basic Physiological Needs, 2 items (e.g., “I cannot take an uninterrupted 30 min mealtime break.”).

Answers are given on a 4-point Likert-type scale, from 1 (Strongly disagree) to 4 (Strongly agree). The total score is calculated by summing the results from the individual scales. The higher the score, the higher the occupational stress levels. The total score was analyzed in the current study. The NOSS yielded a reliability of Cronbach’s α = 0.88.

The Brief Resilience Coping Scale (BRCS) was created by Smith et al. [46]. The scale consists of 6 items (e.g., “I tend to bounce back quickly after hard Times.”) Responses are given on a 5-point Likert-type scale, from 1 (I definitely do not agree) to 5 (I definitely agree). The mean score is the index of resilience. Scores between 1.00 and 2.99 indicate low, while scores between 3.00 and 4.30 are medium, and scores above 4.31 are high resilience. The reliability of the scale measured with Cronbach’s α was α 0.93.

The Intention to Leave Scale measures the employee’s intention to leave the organization and comprises 4 items (e.g., ”I am planning to look for a new job.”) [47]. Answers are given on a 5-point Likert-type scale, from 1 (strongly disagree) to 5 (strongly agree). The mean score indicates the level of intention to leave. The higher the score, the higher the intention to resign. The reliability of the scale measured with Cronbach’s α was α 0.93.

The Job Satisfaction Scale by Zalewska measures general cognitive job satisfaction [48]. It is inspired by Diener et al. Satisfaction with Life Scale, which measures general cognitive life satisfaction [49]. Based on this measure, the Job Satisfaction Scale was created. The items were reformulated such as to concern various aspects of work as a complex phenomenon and required a conscious rating of one’s job based on one’s personal criteria. The scale consists of 5 items (e.g., “In many aspects, my job is close to ideal”). Answers are given on a 7-point Likert-type scale, from 1 (I definitely do not agree) to 7 (I definitely agree). The sum or mean of all responses is the index of job satisfaction. The current study analyzed the mean scores. The higher the score, the higher the job satisfaction. The scale yielded a reliability of Cronbach’s α of 0.85.

Additionally, a demographic questionnaire was included, which asked about the occupation (nurse vs. midwife), age, sex, job experience in years, number of workplaces, weekly number of evening and night shifts, working (or not) at a unit directly dedicated to treating or diagnosing patients with COVID-19, frequency of contact with COVID-19 patients, and having a supervisory or executive position.

The questionnaire also included an attention check item designed to identify and exclude from the analysis those respondents who did not pay appropriate attention. It was placed at the midpoint of the questionnaire and visually resembled the adjacent items (it had the same scale). The item was as follows: “This is an attention check. It is intended to test whether people are giving answers without first reading the questions. Please select: Definitely not true”.

### 2.3. Sample Characteristics

Data collection included 441 participants. The questionnaires were completed in full by 390 people (96.7% women and 3.3% men), aged between 22 and 66 (*M* = 44.48; *SD* = 10.68). The largest group was comprised of people with job experience of over 30 years (29%). Job experience between 26 and 30 years was reported by 16.4% of the sample; between 21 and 25 years by 16.3%. The lowest job experience (1–5 years) was reported by 15.6% of the sample. The sample chiefly consisted of nurses (82.8%). Most of the participants did not occupy executive roles (86.9%). Over half of the respondents had an MA degree (52.8%) and worked at only one location (64.9%). Around 1% of the sample reported having a Ph.D. degree. Almost 7% of the sample was comprised of participants employed in three or more organizations. A total of 40% of respondents had two night shifts per week. Only 17.1% of respondents indicated having no contact or only sporadic contact with COVID-19 patients, though 73.8% of the respondents did not work at a COVID unit.

### 2.4. Data Analysis Methods

Statistical analyses were carried out using the IBM SPSS Statistics 25.0 software. To establish the relationship between the variables, Pearson’s r correlation analysis was carried out. To examine whether occupational stress level is a mediator in the relationship between resilience and job satisfaction or intention to leave, mediation analysis using A. Hayes’ PROCESS 3.6 macro [50]. To establish whether organizational factors are a moderator between resilience and job satisfaction or intention to leave, moderator analysis using A. Hayes’ PROCESS 3.6 macro [50]. (Model 1) was carried out. Both analyses included a bootstrap sampling of 5000 with a 95% confidence interval. Finally, a multiple stepwise linear regression was carried out to establish which variables are predictors of job satisfaction and intention to leave. The significance level was set at α = 0.05.

## 3. Results

### 3.1. Descriptive Statistics

Table 1 shows the basic descriptive statistics together with the Kolmogorov–Smirnov test for the quantitative variables included in the study. The analysis showed that only occupational stress levels assumed a Gaussian distribution, while skewness values for the other variables did not cross the threshold of 1, indicating that the deviation from the normal distribution was not significant [51].

To describe intention to leave, job satisfaction, and occupational stress levels, the results were divided. The division was carried out based on the mean results, such that scores below 1 SD were described as low, scores above 1 SD were described as high, and mean ± 1 SD scores were described as medium. Low job satisfaction was reported by 18.2% (N = 71), medium job satisfaction by 64.9% (N = 253), and high job satisfaction by 16.9% (N = 66) of the sample. Low intention to leave the organization was reported by 16.4 (N = 64), medium intention to resign by 58.7% (N = 229), and high intention by 24.9% (N = 97) of the sample. High occupational stress was reported by 15.6% of the sample, while medium occupational stress by 69.5% of the sample. Low occupational stress was reported by 14.9% of the sample.

### 3.2. Relationships between Resilience, Job Satisfaction, Intention to Leave, and Occupational Stress

To establish the relationships between the measured variables, Pearson’s r correlation analysis was carried out, which showed an average positive correlation between resilience and job satisfaction (*r* = 0.30) and between intention to leave and occupational stress (*r* = 0.46). This means that as resilience increases, so does job satisfaction, while as occupational stress increases, so does the intention to resign. Job satisfaction was negatively and strongly correlated with intention (*r* = −0.59) and with occupational stress (*r* = −0.55), while resilience was weakly and negatively correlated with intention to resign (*r* = −0.29) and occupational stress (*r* = −0.29). The higher the job satisfaction and resilience, the lower the occupational stress and intention to leave. Detailed results of the analyses are shown in Table 2.

### 3.3. Occupational Stress as a Mediator between Resilience and Job Satisfaction

To establish the mediational role of occupational stress in the relationship between resilience and job satisfaction, a mediation analysis using A. Hayes’ PROCESS 3.6 macro (Model 4) was carried out, with 5000 bootstrap sampling [50].

Figure 2 shows the standardized regression coefficients for the relationships between the variables.

The model showed a suitable fit to data *F*(2.387) = 92.81; *p* < 0.001 and it explained 32.4% of the variance in job satisfaction (adj. R^2^ = 0.324).

The analysis showed a significant relationship between resilience and occupational stress (B = −0.17; SE = 0.03; *t*(1.388) = −6.02; *p* < 0.001); as resilience increases, occupational stress decreases. A direct relationship between resilience and job satisfaction was also significant (B = 0.51; SE = 0.08; *t*(1.388) = 6.16; *p* < 0.001); the higher the resilience, the higher the job satisfaction.

The relationship between occupational stress and job satisfaction was also significant when resilience was included (B = −1.48; SE = 0.13; *t*(2.387) = −11.60; *p* < 0.001); the higher the occupational stress, the lower the job satisfaction. A significant relationship between resilience and job satisfaction when occupational stress was included was also observed (B = 0.25; SE = 0.07; *t*(2.387) = 3.44; *p* = 0.001).

To test the significance of the indirect effect on the relationship between resilience and job satisfaction, additional analyses were carried out using the bootstrapping method (with a sampling of 5000). The indirect effect was statistically significant (B = 0.25; BootSE = 0.05; BootLL = 0.15; BootUL = 0.35), which indicates a partial mediation; the relationship between resilience and job satisfaction through occupational stress weakens, but remains statistically significant.

### 3.4. Occupational Stress as a Mediator between Resilience and Intention to Leave

Analogous analyses were carried out to establish the mediational role of occupational stress on the relationship between resilience and turnover intention.

Figure 3 shows the standardized regression coefficients for the relationships between these variables.

The analyzed model also had a suitable fit to data *F*(2.387) = 55.99; *p* < 0.001 and it explained 32.4% of the variation in intention to leave (adj. R^2^ = 0.224).

The analysis showed a significant relationship between resilience and occupational stress (B = −0.17; SE = 0.03; *t*(1.388) = −6.02; *p* < 0.001); the higher the resilience, the lower the occupational stress. A direct relationship between resilience and intention to resign was also significant (B = −0.33; SE = 0.07; *t*(1.388) = −4.63; *p* < 0.001); the higher the resilience, the lower the intention.

The relationship between occupational stress and intention was also significant when resilience was included (B = 1.07; SE = 0.12; *t*(2.387) = 9.26; *p* < 0.001); the higher the occupational stress, the higher the intention to resign. Additionally, a relationship was identified between resilience and intention to leave when including occupational stress (B = −0.15; SE = 0.07; *t*(2.387) = −2.18; *p* = 0.030).

To check the significance of the indirect effect of occupational stress on the relationship between resilience and job satisfaction, additional bootstrapping analyses were carried out (using a sampling of 5000). The indirect effect was statistically significant (B = −0.18; BootSE = 0.04; BootLL = −0.26; BootUL = −0.11), which indicates a partial mediation; the relationship between resilience and intention to leave through occupational stress weakens, but remains statistically significant.

### 3.5. The Moderating Role of Occupational Factors for the Relationship between Resilience and Job Satisfaction

Using A. Hayes’ PROCESS macro (Model 1), moderation analysis was carried out to examine whether organizational factors moderate the relationship between resilience and job satisfaction. Table 3 shows the variable coding system used in the current analysis.

The analysis showed that neither of the organizational variables was a significant moderator of the relationship between resilience and job satisfaction.

### 3.6. The Moderating Role of Organizational Factors for the Relationship between Resilience and Leave

Using A. Hayes’ PROCESS 3.6 macro (Model 1), moderation analysis was carried out to test whether organizational factors moderate the relationship between resilience and intention to leave [50]. The coding of the variables was the same as in Table 3.

The analysis revealed a lack of statistically significant interactions, meaning that neither of the organizational factors under consideration moderated the relationship between resilience and intention.

### 3.7. Organizational Factors, Occupational Stress, and Resilience as Predictors of Job Satisfaction and Intention to Leave

To establish whether any specific organizational factors, occupational stress, or resilience are predictors of job satisfaction or intention to leave, a multiple stepwise linear regression was carried out, with the probability of *F* < 0.05 as an inclusion criterion. The following variables were included in the model as predictors: sex, job experience, education, occupation (nurse vs. midwife), supervisory/executive position, number of workplaces, number of evening and night shifts, working at a COVID-19 unit, resilience, and occupational stress. In the following analysis, the final stage of the model, including the final list of predictors, is presented.

The first analyzed model (Table 4) included job satisfaction as the explained variable. The model had a suitable fit to data, *F*(3.323)= 58.83; *p* < 0.001 and it explained 34.7% of the variance in the explained variable (adj. R^2^ = 0.347). Among the predictors included in the model, the following three were statistically significant: occupational stress, resilience, and number of workplaces. The higher the occupational stress (β = −0.53), the lower the job satisfaction. Resilience (β = 0.17) and number of workplaces (β = 0.11) were positively related to job satisfaction, indicating that the higher the resilience and the higher the number of workplaces, the higher the job satisfaction.

The second analyzed model (Table 5) included the intention to leave as the explained variable. The model had a suitable fit to data, *F*(4.322)= 30.20; *p* < 0.001 and it explained 26.4% of the variance of the explained variable (adj. R^2^ = 0.264). The analysis revealed four statistically significant predictors: occupational stress, job experience, occupation (nurse vs. midwife), and resilience. The higher the occupational stress (β = 0.43), the higher the intention to resign. The longer the job experience (β = −0.11) and the higher the resilience (β = −0.12), the lower the intention. Nurses reported lower intention to leave than midwives.

## 4. Discussion

Thus far, the literature indicated that the specific work environment can be considered a predictor of employee stress. Expanding the knowledge on this point, the current study showed the validity of using this conceptualization to analyze job satisfaction. Moreover, the authors assumed that the specific characteristics of working as a nurse or a midwife during the pandemic period would be reflected in the intention to leave the organization.

Working in healthcare during the pandemic can have very high mental demands. The current study aimed to describe the occupational characteristics of nurses and midwives after about a year of work during the SARS-CoV-19 pandemic. Based on the experiences of work during the pandemic and on the available literature, it was anticipated that higher levels of occupational stress and job dissatisfaction indices would be found in the sample [52]. Descriptive statistics showed, among others, that almost ¼ of the sample reported high intention to leave the organization, and a similar proportion reported low job satisfaction. These results warrant concern. Analyses using a similar measure revealed that close to a half of a sample of Iranian nurses reported medium and high intention to resign; in the current study, this was reported by 83.6% of the sample [53]. A Slovene study comparing the pre-pandemic period to the pandemic period also showed a decrease in job satisfaction in a sample of healthcare professionals, together with a significant experience of individual occupational burnout symptoms in 2020 [54]. It is known that occupational burnout is a reaction to long-term emotional and interpersonal workplace stressors [55].

It seems understandable that, in accordance with the job demands resources theory, work during the pandemic is characterized by low controllability and increased demands from the medical system, which, according to the literature, increases the risk of overwork and negatively impacts the intention to leave, among others [56]. The authors obtained data from employees representing a job environment that experienced rapid and extensive changes. It involved, among others, rapidly implementing new work procedures, relocating patients and staff, structural changes in the organizations and in the procedures of ambulatory care, and the emergence of a new risk of illness and death. Thus, the job demands changed due to the specifics of the situation. Thus, reflections on the relatively greater impact of individual traits or environmental factors on job satisfaction gained importance. In the current study, the personal capabilities of adapting to stressful (and potentially traumatizing) situations have been included as a potential factor with a positive influence.

In the current study, resilience was related to nurses’ job satisfaction during the pandemic. A positive relationship between resilience and job satisfaction among nurses was reported in the literature previously [57]. This is understandable, as resilience allows for sustaining efficient functioning and coping with difficulties. It also allows for accomplishing developmental goals, and it is related to mental health and well-being. Resilient individuals perceive stressful situations as challenges and treat failures as normal outcomes [58]. Resilience allows for regaining well-being after traumatic experiences. Although meta-analyses show that stress has a negative effect on resilience, there are also reasons to consider resilience as a universal resource guaranteeing optimal workplace functioning, irrespective of the pandemic [59]. Studies of nurses during the pandemic show a protective effect of resilience against stress and occupational burnout [60]. In a sample of pediatric residents, high stress levels (related to the uncertainty characterizing this profession) together with low resilience were strongly related to depression and occupational burnout [61]. It seems warranted to use empirical data in workshops dedicated to healthcare professionals. The importance of free access to psychological findings for nurses has been indicated only recently, in the wake of the ongoing pandemic. In light of the data, such workshops should be aimed at healthcare workers regardless of the presence or absence of pandemic risk.

The current analyses allowed for establishing that the higher the job satisfaction, the lower the intention to leave and that this relationship was highly statistically significant. This is consistent with studies on nurses, including the Polish studies carried out a year prior to the pandemic [4]. Job satisfaction refers to the employee’s expectations being met, and it may be an effect of occupational stress [62]. In turn, occupational stress negatively impacts job motivation and effectiveness [63]. If accruing stressful experiences progress into occupational burnout, they also impact the intention to resign [64]. Occupational stress has a documented effect on nurses and other healthcare professionals. They experience stress-related health problems at a higher rate than other occupational groups [65]. Thus, occupational stress is indicated as the most frequent source of occupational burnout, and burnout as the reason for quitting nursing [66]. In a sample of palliative nurses, burnout mediated the relationship between resilience and secondary traumatic stress resulting from workplace experiences [67]. Medical personnel’s occupational stress weakens the relationship between job satisfaction and resilience and between resilience and intention to resign.

Surprisingly, neither of the organizational variables considered in the current study was a statistically significant moderator of the relationship between resilience and job satisfaction, and intention to leave. The current study did not show a significant effect of direct contact with COVID-19 patients. Other results show that, aside from occupational stress, the risk of infection strongly impacts the intention to leave among nurses [68]. In contrast to the current results, a study from April 2020 showed that contact with COVID-19 patients has a significant effect on resilience [69]. Other studies also showed that front-line nurses during the pandemic experienced fear and anxiety, but also empathy and compassion [70]. Fear of infection and intention to leave the organization were related to resilience during the pandemic in a sample of nurses from four countries (Japan, Republic of Korea, Republic of Turkey, and the United States) [71].

In the predictive models for job satisfaction tested in the current study, the organizational factor of the number of workplaces was significant (positively related), while job experience was a positive, negative predictor of intention to resign. Intention to leave is the employee’s declaration, which may (though it does not have to) mean their reduced work engagement. It can be assumed that, in the employee’s estimation, their current workplace is not sufficiently attractive [72]. The variable of job experience (in years) has frequently been considered in empirical studies. Thus, in a sample of nurses, it correlated positively with resilience and negatively with the symptoms of secondary traumatic stress [73]. In a sample of mental health care professionals, a negative correlation between job experience and emotional exhaustion, being a factor of occupational burnout, was observed [74]. Simultaneously, first responder team nurses’ job experience was positively correlated with the intention to leave [75].

The number of workplaces frequented by one employee cannot be ignored in the context of job satisfaction. Polish healthcare professionals working in multiple workplaces have thus far been described as an undesirable phenomenon. Nursing resources per 1000 citizens in Poland are some of the lowest among the OECD (Organization for Economic Cooperation and Development) nations, and it is estimated that the proportions of employed nurses and midwives will continue to lower until 2030 [76].

A 2021 empirical analysis of this phenomenon showed that 44% worked in two workplaces, working over 160 h per month (39%). The most frequent reason for nurses working at multiple workplaces (93.3% of the sample) is financial concerns [21]. Simultaneously, there are reports showing that multiple workplaces can have benefits other than financial ones, including greater job satisfaction and learning new skills [77].

## 5. Conclusions

Based on the results of the current study as well as the available literature, it has to be indicated that:Intention to leave the organization was declared by almost ¼ of the sample; medical personnel more often reported low than high job satisfaction;Resilience plays a role in shaping job satisfaction and intention to leave among medical professionals;Occupational stress weakens the relationship between resilience and intention to leave and between job satisfaction and resilience.

For job satisfaction and intention to leave, organizational factors such as job experience and number of workplaces were statistically significant in predictive models.

## 6. Limitations of the Study

The current study has several limitations. Above all, due to the public health regulations put in place, it was impossible to carry out the study in the traditional pen-and-paper format. As a result, online data collection prevented direct contact with the participants. Researchers’ physical presence at the studied organizations would certainly expand the perspective on occupational problems faced by medical professionals during the pandemic.

The mean age of the current sample was 43, whereas the mean age of nurses in Poland is 52 [78]. Around 27% of them are above 60 years of age. Thus, nursing is indicated as an aging profession. Close to 70 thousand nurses and midwives have already reached retirement age and could retire at any moment. It is likely that few of them had the necessary digital literacy skills required to fill out an online questionnaire, which represents a limitation of the current study.

Polish medical professionals’ functioning during the pandemic is a new area of study. Thus, it was important to gather as much data as possible. This is especially more so due to the uncertainty regarding the course of the pandemic and the current staffing problems. In light of the statistics showing that medical professionals are significantly overworked during the pandemic, other variables that may have played a role have not been included. These are, for example, coping styles, health, including mental health, and organizational support. They have been widely studied in workplace and organizational psychology [35]. Additionally, the current study did not account for chronic fatigue and overwork as factors that may have had a potentially significant impact on the relationship between resilience and intention to leave or job satisfaction. The authors decided that the questionnaires should not be too long or taxing to fill out due to both practical and ethical considerations.

Midwives also represented a minority in the current sample. Evidence regarding this group in comparison to nurses is still lacking. The chief practical difference is that, with more people requiring nursing services, the need for midwives is different. The birthrate in Poland is decreasing yearly [79].

## 7. Future Research

Further studies should consider the impact of both short- and long-term changes to nurses’ and midwives’ work during the pandemic. In many countries, medical professionals working directly with COVID-19 patients have been given raises. In Poland, they were 50% in 2020 and 100% in 2021, respectively. The financial factor—as the current study showed—is an important element shaping job attitudes (including job dissatisfaction) in the Polish healthcare system. Moreover, in Poland, the National Labor Union of Nurses and Midwives, the largest labor organization of medical professionals, began a strike in September 2021. At the moment of submitting the current article, the strike is ongoing. The strikers’ demands include quicker increases in budget spending on the healthcare system and raising healthcare workers’ salaries to the mean OECD and EU levels. This not only reflects medical professionals’ frustration but also offers a unique opportunity for empirical analyses. For example, it could be examined who engages in striking and collective action and what effects it has on their job satisfaction or intention [80].

Further studies on this topic should also consider the fact that in Poland, around 70 thousand people are educated as nurses and midwives but do not work in these professions [81]. Reaching this population may be difficult, but not impossible. On the other hand, studies could examine the factors that determined their career change. Data from other studies point to such factors as the negative impact of an unsupportive work environment [82]. In this context, considering basic psychological needs in terms of frustration rather than lack of satisfaction could provide a more adequate understanding of the harmful effects that psychological distress could have in the workplace [83].

The current analysis focused on examining intention to leave and job satisfaction levels among medical professionals. However, identifying methods to increase positive job appraisals is as important as identifying the causes of negative job appraisals [24]. The literature points to a positive influence of intervention programs aimed at developing resilience, which results in lower stress and lower burnout symptoms [84,85]. It seems necessary to reanalyze systemic solutions dedicated to healthcare workers during the pandemic across many countries. Moreover, retrospective studies could aim at gathering self-report data from medical professionals. For example, they could examine what helped them sustain a goal-oriented motivation despite the experienced difficulties, how much did the pandemic exacerbate the pre-existing occupational problems, and which problems emerged as new ones.

## Figures and Tables

**Figure 1 ijerph-19-06826-f001:**
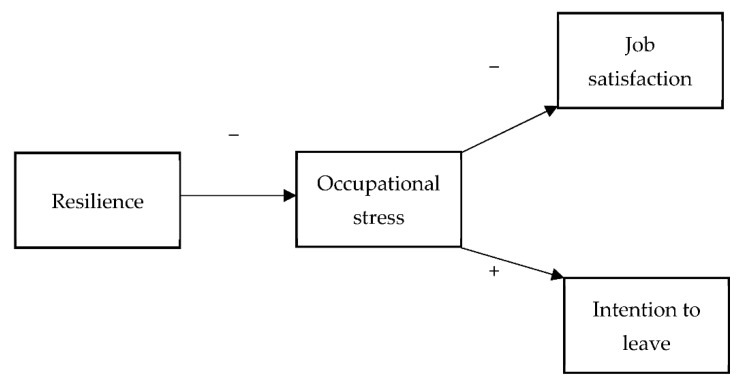
Model tested.

**Figure 2 ijerph-19-06826-f002:**
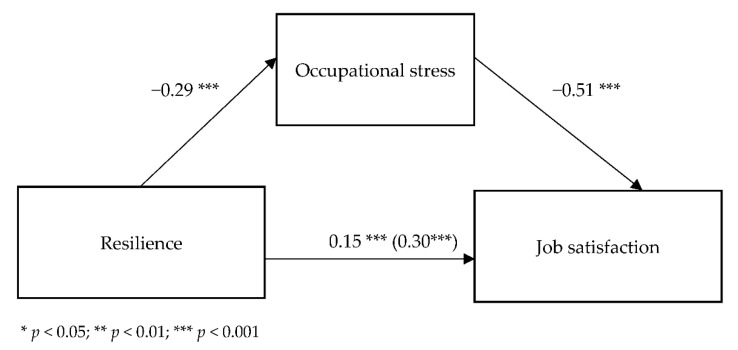
Standardized regression coefficients for the mediational model of occupational stress between resilience and job satisfaction.

**Figure 3 ijerph-19-06826-f003:**
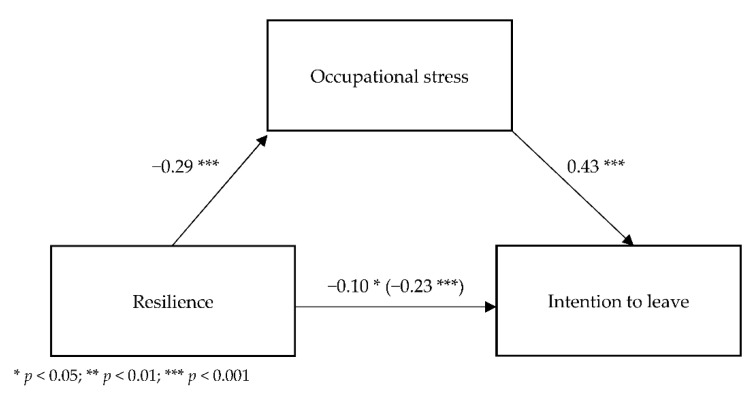
Standardized regression coefficients for the mediational model of NOSS in the relationship between resilience and intention to leave.

**Table 1 ijerph-19-06826-t001:** Descriptive statistics and the Kolmogorov–Smirnov test of distribution normality.

	M	Me	SD	Sk.	Kurt.	Min.	Max.	D	*p*
Job satisfaction	3.81	3.80	1.29	−0.13	−0.67	1.00	6.80	0.07	<0.001
Resilience	3.02	3.00	0.76	−0.06	−0.58	1.00	5.00	0.07	<0.001
Intention to leave	2.83	3.00	1.10	0.08	−0.87	1.00	5.00	0.13	<0.001
Occupational stress	2.87	2.86	0.44	−0.20	0.13	1.29	3.95	0.04	0.077

**Table 2 ijerph-19-06826-t002:** Pearson’s r correlations between resilience, job satisfaction, intention to leave, and occupational stress.

	1	2	3	4
1. Job satisfaction	1			
2. Resilience	0.30 **	1		
3. Intention to leave	−0.59 **	−0.23 **	1	
4. Occupational stress	−0.55 **	−0.29 **	0.46 **	1

** *p* < 0.01.

**Table 3 ijerph-19-06826-t003:** Qualitative variable coding.

Variable	Code	Description
Workplace	W1	Having a single workplace
	W2	Working at two places
	0—Reference category	Working at three or more places
Nightshifts	W1	One nightshift
	W2	Two nightshifts
	0—Reference category	Three or more nightshifts
Working at a COVID-19 unit	0—Reference category	No
	1	Yes
Supervisory/executive position	0—Reference category	No
	1	Yes

**Table 4 ijerph-19-06826-t004:** Multiple linear regression model explaining job satisfaction.

						95% CI	
	B	SE	Β	T	*p*	LL	UL
(Constant)	7.09	0.53		13.36	<0.001	6.04	8.13
Occupational stress	−1.57	0.14	−0.53	−10.99	<0.001	−1.85	−1.29
Resilience	0.30	0.08	0.17	3.64	<0.001	0.13	0.45
Number of workplaces	0.23	0.09	0.11	2.46	0.014	0.05	0.42

**Table 5 ijerph-19-06826-t005:** Multiple linear regression model explaining intention to leave.

						95% CI	
	B	SE	Β	t	*p*	LL	UL
(Constant)	1.01	0.52		1.94	0.053	−0.01	2.04
Occupational stress	1.09	0.13	0.43	8.66	<0.001	0.84	1.34
Job experience	−0.09	0.02	−0.18	−3.74	<0.001	−0.14	−0.04
Occupation	−0.33	0.14	−0.11	−2.39	0.017	−0.60	−0.06
Resilience	−0.17	0.07	−0.12	−2.38	0.018	−0.32	−0.03

## Data Availability

Data available on request due to restrictions privacy.

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
