# Peer review of "Resilience, Occupational Stress, Job Satisfaction, and Intention to Leave the Organization among Nurses and Midwives during the COVID-19 Pandemic"

_ijerph, 2022, doi:10.3390/ijerph19116826_

Round 1

Reviewer 1 Report

The manuscript ironically underestimates its methodological potential. In fact, in the title and in the abstract there is no mention of moderated mediation and also of multiple regression, on the contrary the abstract is unstructured and not very appealing, then the results come in great depth! I suggest an overall reshaping of your work, underlining its strengths, and re-evaluating the English term "turnover intention" because it does not seem to me to convey the concept you are analyzing.

L12 Remove” There is a high demand for mid-level professionals in the healthcare sector such as nurses 12 and midwives.” Mid-level professionals is misleading..

L14 remove 390

There is no time frame

L18 Enter here the result of inclusion of total nurses and total midwives... (https://www.researchgate.net/publication/353522424_Checklist_for_Reporting_Of_Survey_Studies_CROSS )

L23 “burden”

L27-30 Reference missing

L40-44 Reference missing

L73-143 These 3 sub-paragraphs should be moved to methods because are outcome measures: Turnover intention, job satisfaction, resilience .. Are you sure about the term "turnover intention"? It would not be better to intend to resign?

L162 Design.. Cross-sectional survey study? Where, when?

Criteria for inclusion and exclusion of participants totally missing

In the results it is superfluous to provide a tabulation of the normality study, you must describe the sample .. the number .. the hours of work complied with in the pandemic .. years of experience ... public, private assignment ... supervised or not .. days of holidays? I do not know

L291 pointed out in the title that there was "a moderated mediation and multiple regression analysis"

L453 I suggest a statement and ref. as follows: “In this context, considering basic psychological needs in terms of frustration, rather than lack of satisfaction, could provide a more adequate understanding of the harmful effects that psychological distress could have in the workplace.” (ref: https://www.mdpi.com/1660-4601/18/18/9676/htm )

Author Response

Dear Reviewers, Dear Professors,

Thank you for your suggestions regarding our article. We have incorporated them into the revision. Below, we include details about the changes we have made. The most important changes are marked in color. Deleted text has been crossed out.

Review 1

In accordance with the Reviewer’s suggestions, we have revised the abstract.

The phrase “turnover intention” was changed to “intention to leave the organisation,” which is used interchangeably in the text with “intention to resign.” Also according with the suggestions, we deleted other fragments and phrases in the text which caused confusion, as indicated in the review. A sentence, together with a reference citation, was also added, in accordance with the Reviewer’s suggestion.

The References have been checked and ordered.

Criteria for inclusion and exclusion of the participants were also added in the text.

Review 2

A graphical model based on the Job-Demands Resources theory was added in the Introduction.

The description of the moderation results was reduced to a minimum.

The study yielded a total of 4 questionnaire-based variables and 9 sociodemographic variables. Due to their number, we decided against including all of the data in table form.

The discussion of the results was expanded based on the JD-R model as well as on the practical implications of the results and the literature review on the role of resilience in work.

Academic Editor Comments

The title and keywords were changed as per the suggestions.

Yours sincerely,

Authors

Reviewer 2 Report

Thank you for giving me the chance to review the manuscript ijerph-1680923 titled " Resilience, occupational stress, job satisfaction and turnover intention among nurses and midwives during the SARS-CoV-2 pandemic". The authors investigated the turnover intention and job satisfaction of a sample of Polish nurses and midwives by taking into account a series of explanatory variables, e.g., occupational stress and resilience.

I read the paper with great interest, and I believe that this project may be able to make a considerable contribution to the literature if the authors are able to address the following points:

(1) The study is depicted as rather an explanatory one while it tests hypothetical (indirect/mediation) models (amongst others). Hence, the introduction is focused on offering state of the art regarding the importance of the topic and the alarmingly turnover phenomenon among this specific occupational population (especially during the COVID-19 pandemic), but it is missing an integrative theoretical framework for the investigated variables. None of the studied factors are new to the literature (e.g., occupational stress, resilience, turnover intention, and job satisfaction), thus, all of them can be coherently integrated into a theoretical model. Therefore, my first major suggestion would be to reshape the introduction in order to elaborate on the theoretical arguments that lead you to those mediation models and each path of the models to be prefigurated through a hypothesis. Moreover, since the two models differ only regarding the outcome, these could be integrated into a single overall, multivariate model (only that you will have to employ Structural Equation Modeling for testing it). As a theoretical framework, you can use the Job-Demands Resources theory (JD-R; Bakker & Demerouti, 2017). Also, I suggest you graphically depict the model at the end of the introduction.

Bakker, A. B., & Demerouti, E. (2017). Job demands-resources theory: Taking stock and looking forward. Journal of Occupational Health Psychology, 22(3), 273–285. http://dx.doi.org/10.1037/ocp0000056

(2) Furthermore, from an analysis standpoint, you begin by describing the distribution of each variable. If the frequencies are normally distributed (following a Gaussian shape), then, indeed, approx. 68.2% of the observations will fall between +/- 1 SD, and approx. 15.9% to the right and left of 1SD. It would be better to compare these data to other reports on the same occupation, or at least use a reference the scale based on which the assessment took place (e.g., if 4 meant “Agree” on the Likert scale for assessing the turnover intention, they we may what to report the percentage of participants who had an average level equal or greater than that).

(3) Also analysis-wise, you tested a series of moderation effects. From what I can see, none of the interactions were statistically significant. Maybe it would be better to discard these analyses from the manuscript and focus exclusively on the mediation models. Moreover, the multiple regressions you introduced, in the end, may be the first models to begin the results section with. Start with these, where the workplace characteristics (tested as moderators) which correlate with the outcomes may be treated as controls, and continue with testing the indirect effects (mediation models

(4) Please provide a full correlation matrix including all studied variables and the main demographics (i.e., age and gender)

(5) The discussions should be reshaped based on the modified intervention and focused more in-depth on the theoretical contribution of the study. Also, the practical implications could be improved. The role played by resilience could be discussed from an interventional perspective (what insights could be transferred from the current study to the literature on occupational health interventions).

I hope my comments will be helpful to you in revising the article.

Author Response

(The authors gave the same response as above.)
